# Computational Thinking and Educational Robotics Integrated into Project-Based Learning

**DOI:** 10.3390/s22103746

**Published:** 2022-05-14

**Authors:** Albert Valls Pou, Xavi Canaleta, David Fonseca

**Affiliations:** Research Group on Technology Enhanced Learning (GRETEL), La Salle, Ramon Llull University, 08022 Barcelona, Spain; xavier.canaleta@salle.url.edu (X.C.); david.fonseca@salle.url.edu (D.F.)

**Keywords:** computational thinking, educational robotics, project-based learning, STEAM

## Abstract

In the context of the science, technology, engineering, arts and mathematics disciplines in education, subjects tend to use contextualized activities or projects. Educational robotics and computational thinking both have the potential to become subjects in their own right, though not all educational programs yet offer these. Despite the use of technology and programming platforms being widespread, it is not common practice to integrate computational thinking and educational robotics into the official curriculum in secondary education. That is why this paper continues an initial project of integrating computational thinking and educational robotics into a secondary school in Barcelona, Spain. This study presents a project-based learning approach where the main focus is the development of skills related to science, technology, engineering, arts and mathematics and the acquisition of computational thinking knowledge in the second year of pupils’ studies using a block-based programming environment. The study develops several sessions in the context of project-based learning, with students using the block-programming platform Scratch^TM^. During these sessions and in small-group workshops, students will expand their knowledge of computational thinking and develop 21st-century skills. We demonstrate the superior improvement of these concepts and skills compared to other educational methodologies.

## 1. Introduction

The use of educational robotics (ER) as a pedagogical resource is a current topic of research as this is becoming quite well integrated into schools and high schools [1]. Similarly, computational thinking (CT) [2] is being introduced in the pre-university educational stages [3]. ER and CT are being introduced either as separate subjects themselves or as part of learning within the environment of science, technology, engineering or mathematics (STEM subjects) [4,5,6,7], without neglecting creativity and Art-STEAM as an indispensable part of students’ development [8,9,10]. Educational robotics and computational thinking, directly or indirectly, evoke the use of technology based on ideas such as experimentation, design or programming using computers [11]. 

Virtual, software-based programming has evolved into accessible, tangible programming [12]. This development has followed the pretexts of constructivism, either by considering the student as the builder of their own knowledge [13] or considering the student as a result of social and cultural interaction with the environment [14]. Recently, tangible programming has been applied to develop proposals for learning techniques [15,16].

Educational robotics covers many technological aspects, but is basically focused on the use of educational robots, basic programming knowledge and problem-solving in the engineering environment [17]. Computational thinking, meanwhile, at a pre-university level, focuses on laying the foundations for future programmers and code developers and enhancing the skills needed for the 21st century [18], Such teaching aims to engage students in activities or projects where they work on and develop logical thinking [1,7,19,20].

Concepts related to STEAM subjects, robotics and computing are important if we are to meet the growing demand for professionals in related fields given the current lack of technology professionals [21]. Accordingly, they should motivate students to continue with these disciplines [22].

Integrating STEAM subjects into educational curricula represents a starting point for these technical disciplines to be strengthened [4]. This integration into curricula can be done through technological platforms, such as robotic kits and programming tools, which can support learning about educational robotics and computational thinking, and using different methodologies [1], too, especially those that target the early ages [23]. At the same time, using ICT to promote the development of a person in society is often promoted [24]. For these reasons, the connection between the world of education and social reality sets the basis of learning in schools. In that context, STEM or STEAM learning is increasingly based on project-based learning (PBL) [25].

The application and development of PBL as a methodology not only increases motivation and learning but also promotes the development of the skills needed for the 21st century [26,27,28,29]. One of the aims is to learn computational thinking through the use of technological programming platforms in the field of education, in a global and interconnected framework such as PBL.

Although not all education systems incorporate this content into the official curriculum at the secondary level, most governments, organizations or teachers encourage initiatives so that students can participate in events where the skills and knowledge of technical disciplines are developed. In recent years, events such as the FIRST^®^ LEGO^®^ League, World Robotics Olympiad^®^, RoboCup^®^ and VEX^®^ Robotics Competition have encouraged increased participation based on motivation and learning from STEAM disciplines. Although these competitions are not based on the official curricula of some countries or states (as is the case in Spain), they do have common features of concepts and skills used in robotics and computational thinking [30,31].

Further to this, numerous examples and studies integrate the disciplines of robotics and computational thinking at different stages of education. Of note are the studies on introducing the use of computational thinking, with a specific corresponding curriculum in early childhood education, including robots and tangible interface programming [32], as well as learning computational languages using programmable blocks [33]. In primary education, it is necessary to emphasize the use of platforms such as LEGO^®^ WeDo^®^ [34] or LEGO^®^ Mindstorms^®^ [35] and block programming using Scratch™ [36]. Previous studies focused on applications from other platforms and questioned the relationship between robot-man and the goal of using robots in different areas of learning, such as language, science and technology [37].

It is in secondary education where the learning and development of STEAM subjects has mainly developed [38], along with the application and use of robotics to develop a range of 21st-century skills and aptitudes [39]. In many cases, activities are developed based on the concept of powerful ideas and the use of computers and robots to promote socialization and learning [11].

The use of technology platforms such as LEGO^®^ Mindstorms^®^ or similar, and block programming such as Scratch™, encourages students to learn programming, educational robotics and computational languages in educational settings [36]. These platforms, when applied at the secondary level, encourage creativity in contextualized environments [40]. Both platforms have in common the concepts of constructivism and constructionism. As a common feature, both educational theories evoke the development of knowledge from previous ideas, allowing students to be the creators of their own cognitive tools as well as their external realities. In the same way that a scientist’s vision changes depending on the environment, students’ knowledge and creativeness are influenced by the learning environment [41]. The most significant difference is at a deeper level: while Piaget’s interest in constructivism lies primarily in building internal stability, through conserving and reorganizing assets for learning, Papert’s constructivism is more interested in the dynamics of change or the discovery of novelty [41].

Today, teacher training is a key part of the future of education and is of great importance within universities. In this training, one of the models that are used and widely accepted is the TPACK model [42], which has been significantly important in the technological field. Technology (T) with a pedagogical use (P) and that is focused on enhancing knowledge (K) without neglecting the teacher’s environment (C) represents a good theoretical framework to encompass the use of new technologies in the classroom.

If they are trained to do so, teachers may be indispensable for applying and promoting the use of technological platforms, thereby increasing students’ knowledge and learning of certain concepts related to computer science or computational thinking. The TPACK model then goes slightly beyond the training environment of future teachers and considers the profile and environment of the student, as can be investigated through project-based learning, which represents an approach to understanding the social and real-world environments of the student. None of these learning methodologies or models are far removed from Piaget’s constructivism [13], but they are substantially different if one considers the influence of the hyperconnected society that exists in the 21st century. This connectivity affects many areas, both in our social and daily lives, and the acquisition and dissemination of knowledge. Connectivism [42] has acted as an enhancer for obtaining knowledge and information.

From the beginning, the application of the TPACK model has focused on the adult environment of teacher training [43], focusing on the learning process [44]. If only this circumstance is taken into account, it can be said that the TPACK model is designed “for teachers”. Yet, given that it is a model applied to training and all related aspects (T, P, C, K), little potential is gained and taken away from it if it is only applied to teacher training, as most students are not in the postgraduate or master’s stages but in compulsory education (mainly primary and secondary), when they are not adults but children. To implement this approach more widely, project-based learning has great potential. Working within a PBL environment meets many of the requirements for developing 21st-century skills [26] and acquiring knowledge in a cross-cutting and interconnected way, as students receive technological training in which their own environment is taken into account. 

In this article, TPACK guides the conception and elaboration of the proposal. We use the intrinsic characteristics and concepts in this model to support program design. In doing so, we propose designing sessions based on the development of computational thinking, educational robotics and 21st-century skills with a STEM or STEAM objective. We use the TPACK model to systemize the design of training activities and assessments, keeping in mind those three strands of knowledge and their interrelationships.

First, we look at the curriculum (C) that may be appropriate to teach, followed by the methodologies that befit the teaching of this curricular knowledge (PC), in particular, project-based learning. Finally, we think about the elements of technology (T) that may facilitate this methodology (P), to develop the teaching of the curriculum (C) and promote its effectiveness and efficiency (TPC).

With all these factors, we considered how certain basic aspects in the training of students can be strengthened and improved to support their learning of computational thinking, educational robotics and the necessary life skills for the 21st century. We defined two research questions:

R1: Can project-based learning improve students’ skills? 

R2: Does using a visual programming platform facilitate learning programming concepts? 

To answer these questions, the authors designed, implemented and evaluated a corresponding research study, as will be described in the remainder of the article.

## 2. Literature Review 

### 2.1. Computational Thinking

Computational thinking involves solving problems, designing systems and understanding human behavior by drawing on the concepts fundamental to computer science [2]. Therefore, the inclusion of CT in the school curriculum is a fundamental aim in the development of resources and tools for solving problems in STEAM subjects. STEAM best-practice implementations are currently the focus of great interest, not only because of how they are proving useful in improving student motivation in the processes involved but also as a strategy for addressing diversity issues and improving the STEAM vocations of students affected by such gaps in representation [45,46]. 

Often, CT is used in schools when working with basic skills such as reading, writing and arithmetic [2]. However, the inclusion of CT is not really a subject in itself; instead, it is usually linked to technology or robotics. One of the reasons for this is that there has been little development in the dimensions or concepts that should be worked on within the discipline of CT. To solve that, Brennan and Resnick suggest the following key dimensions of the computational thinking framework [47]: computational concepts, practices and perspectives. 

They identified seven computational concepts, separated into four computational practices and three computational perspectives:


*Computational concepts*


Sequences: When programming is very important, the activity or task is expressed as a sequence of individual steps or commands that can be performed by the computer. They must be followed gradually and in the correct order.Loops: A tool that allows us to run multiple sequences and run sequences indefinitely or until some condition that ends the loop is met.Events: An important element of other interactive components and may be used as a trigger for a sequence. As an example, we can take the start button, which can trigger music playback, change the setting or cause the movement of an object.Parallelism: The implementation of several sequences simultaneously. There are two types of parallelism: between sprites and within a single facility. The former means that several sequences are implemented in the same object. With the latter, if each object has a series of different actions that are triggered by the same condition—when the start button is activated in this case—parallelism is used within a sprite.Conditionals: This is the ability to make decisions or take actions based on certain conditions. It serves to determine the conditions under which a sequence takes place.Operators: Provide support for mathematical and logical expressions and strings that allow us to perform mathematical operations (addition, subtraction, division, multiplication, function, etc.) and operations with strings (grouping, a length of the string, etc.)Data: This involves storing, retrieving and updating values. We can use variables, assign the value of a number or string and create lists that may contain a set of numbers.


*Computational practices*


Incremental and iterative: Designing a project is not a clean, sequential process of first identifying a concept, then developing a design plan and implementing it in code format. It is an adaptive process in which the plan can change in response to the approach when searching for a solution in small steps. This can be described as iterative cycles of imagining and constructing: developing a little, trying it out and then developing it more based on your experiences and new ideas.Testing and debugging: Designers must develop strategies to deal with and anticipate problems as things rarely work out as imagined. A fundamental practice of programmers is testing and debugging, techniques that were developed through trial and error, transfer of other activities or with the support of colleagues.Reusing and remixing: Relying on other people’s work is a long-standing practice in programming and has only been amplified by networking technologies that provide access to a wide range of other people’s projects to reuse and remix. Reusing and remixing support the development of code-reading skills and help you find ideas and code to build on, which allows you to create much more complex functions than you could have created on your own.Abstracting and modularizing: Constructing something large by assembling smaller pieces is an important practice for all design and problem-solving. Designers use multilevel abstraction and modularization, right from the initial work of conceptualizing the problem to translating the concept into individual sprites and stacks of code. The modularization of the behavior of the object makes it easier to assess (try/debug) the different parts of the project or problem.


*Computational perspectives*


In a dimension that is not captured by the framework of concepts and practices, perspectives are described by designers and programmers as the evolution of their understanding of themselves, their relationships with others and the technological world around them. 

Expressing: People spend time surrounded by interactive media as simple consumers, performing activities that are important to use technology, but which are not as sufficiently developed in most people as they are in a computational thinker. A computational thinker sees computing as more than something to consume; computing is something that can be used for personal design and expression and is seen as a means to create and express one’s own ideas.Connecting: Creativity and learning are deeply social practices, and so designing computer media is surprisingly enriched by interactions with other people. This fact has been observed in the wide variety of ways in which individual creative practice has benefited from access to others through face-to-face or online interactions. Young developers have described the power of accessing new people, projects and perspectives through these networks, a change of perspective that is succinctly expressed as, “I can do different things when I have access to others.” Creating with other people allows them to do more than they can handle on their own, whether that occurs through answering questions on online forums, studying and mixing with others or establishing intentional partnerships and collaborations. When creating for others, you experience the value of an authentic audience. Whether it is when entertaining, equipping, involving or educating others, you may appreciate that others are getting involved and appreciate their creations.Questioning: Everyday life is increasingly regulated by complex technologies that most people do not understand or believe can have much influence on [48]. With the computational perspective of questioning, we look for indicators to avoid feeling this disconnection between the technologies around us and our ability to negotiate the realities of the technological world. Young developers should feel empowered to ask questions about and with technology: “I can (use computing to) ask questions to make sense of (computational aspects of) the world.” Questioning involves curiosity about that which is taken for granted, and in some cases, answering that question through design and development.

Although during our research we were focused on computational concepts and perspectives, the literature introduced us to concepts, practices and perspectives of CT, all of which were considered. 

Computational concepts such as sequences, loops and conditionals were identified as basic concepts for young learners [49]. A high percentage of students in fourth- to sixth-grade (aged 9–12) can complete sequence-related tasks [50]. The conditional concept was conceived by students, but not fully, particularly when the conditional block was combined with other blocks such as when they were nested within a forever loop [51]. One of the most difficult concepts for students to grasp in these stages was repetition (loops) because students may not have known the condition and procedures needed to complete a loop [52]. All these studies were developed for primary education, especially for students between the ages of 9 and 12, and they suggest that it cannot be assumed students’ understanding of CT concepts will necessarily improve when using a block-based programming environment. However, this research focused on students aged 12–14 who had already worked with a block-based programming environment [40] and developed skills and concepts of computational thinking that could support improved results versus those achieved in primary education.

Computational practice was also analyzed as it has been found that when framing computational thinking only around the concepts, other elements of learning and the participation of designers were insufficiently represented [47]. One of the most similar ideas to other forms of learning is that computational practices focus on the processes of thinking and learning, going beyond the question of what you are learning to how you are learning. CT practice, in particular, raises more doubts than CT concepts, and questions arise such as the following: How can a valid assessment tool be designed for CT practices? Can algorithmic thinking, abstraction, testing and debugging, as well as reusing and remixing, be used in the fourth- to sixth-grade levels? What do these assessment tasks tell us about the individual student? [45]. Previous work in the STEAM high school disciplines has shown improved pre-test and post-test scores based on computational thinking practices [53].

The third point of view in CT is perspectives, which according to Brennan and Resnick [47], focus on the evolution of students’ understanding of themselves, their relationships with others and the technological world around them. These three aspects, for professionals, relate to the interactions that developers have with one another, especially in terms of how to express themselves, contribute and ask peers for support. Few studies have investigated whether this is the same at the secondary level; instead, other researchers have focused on early years learning [32]. Such studies in other stages of education have developed tools to observe these concepts more objectively, thereby defining competences such as communication, creativity and collaboration [54], which are similar to the definition of perspectives from Brennan and Resnick [47].

Previous studies tended to define certain skills related to CT [31] as the capacity for abstraction or modularization, but following Brennan and Resnick [47], these definitions coincide more with CT practices than with perspectives.

Overall, CT has been analyzed in many disciplines and related to other types of thinking, with attempts made to define its characteristics, the relationship between computer science and programming, the most appropriate tools for developing it and the skills connected to CT [55].

### 2.2. Scratch™

Scratch™ is one of the most popular and widely used block-based programming platforms. From the conception of the main idea [56] to the post-development [36] or explanation of the environment [57], it has been used to develop and enhance many CT concepts. 

The literature incorporates many studies, workshops and works related to Scratch™. In primary education, studies have focused on tutorials on how to teach and learn Scratch™ [58], computer games [59], maze development [60] and evaluation programming concepts [61].

In many other cases, researchers have attempted to establish how many blocks to use in creative projects, either in science projects in fifth grade [62], after-school computer clubs [63] or summer camps during middle school [64].

Regardless of this platform’s uses, the main goal of Scratch™ is to program and thus learn the concepts, practices and perspectives of computational thinking [47].

Scratch™, like any other programming platform, requires basic learning by students. If students are relatively young and new to programming (such as the fourth grade), the main goal of the learning courses is to introduce the core features of Scratch™ and CT through content and programming activities that are largely introductory to students [65]. Other projects demonstrate how Scratch™ can be used with young children aged 8–9 years old, who will learn programming concepts through an introduction to game-making [66].

Learning and using Scratch™ is based on students’ prior knowledge, which can come from any other discipline or subject [67]. Just as constructivism encourages the student to be the protagonist of their own learning [13] and constructionism proposes tangible learning elements [15], Scratch™ has emerged as a platform that encourages self-learning based on elements with which the student can create, imagine, design, modify... and learn computational thinking.

### 2.3. Project-Based Learning

Both PBL and inquiry-based learning are active learning methods based on the philosophy of John Dewey, who believed that education should arouse students’ curiosity [68].

Sahin [25] identifies similarities between these two methods if we consider that project-based learning is an educational approach that uses student-centered research processes to develop a product that has real-life connections and applications. Specifically, PBL contains research-based tasks that help students develop important content from the technological, social and core curriculum. PBL was also explored in a special case study [69] that concluded STEAM PBL and inquiry-based learning go hand-in-hand in terms of instructions focused on students’ learning.

However, the differences are significant because with the inquiry-based learning method, the questions and curiosity of students are central to the curriculum. Students are encouraged to ask questions, conduct research on topics that interest them and make their own discoveries [70]. Inquiry-based learning begins with a question followed by research. This research may involve the collection of data and the development of new knowledge, and at the end of this process, students reflect on their new knowledge [71].

Instead, STEM PBL starts with the final product in mind. Students are introduced to PBL through a well-defined outcome that conveys clearly defined expectations and limitations for completing the task [25]. Development through STEM PBL differs from inquiry-based learning in the emphasis on the construction of artifacts or objects by students to represent what they have learned.

Another outstanding feature of PBL in STEM projects is that it is usually unstructured. This is because PBL normally works in small groups of students where each group has to organize its own work, materials and individual tasks, as well as manage its own time [72].

Thus, in STEM project-based learning, students take charge of their own learning and develop collaborative skills. At times, the largely unstructured nature of project-based learning can make PBL classrooms seem disorganized or out of control.

One of the key factors in the successful use of PBL in STEM or STEAM disciplines is its interdisciplinary nature. That is why the use of these methodologies increasingly influences the current teaching of STEAM subjects, computational thinking or educational robotics. Today, there is a desire to create a theoretical framework for designing STEM projects [69] or an integrated STEM approach [73], as well as enhancing informal education and community collaboration through engineering [74].

## 3. Course Design and Methodology

### 3.1. Background

The style of teaching may change and evolve as a result of the new realities and potential of the educational sector (digital natives, new technologies, etc.) [75,76]. The La Salle Educational Mission Assembly resolved in December 2018 (AMEL 2018) to create and execute a new model in all of its centers for all educational levels, from kindergarten to university. The ‘New Learning Context’ (NLC) is the term given to this new approach, which has been in use in Spain for the past two years [77,78].

There are 104 centers engaged in this deployment, including two university colleges, La Salle Campus Barcelona (Ramon Llull University, Barcelona, Spain) and La Salle Campus Madrid (Universidad Complutense de Madrid, Madrid, Spain). The NLC model is founded on five pedagogical principles (interiority, mind-body-movement, thought construction, self-regulated behavior and the social dimension of learning) [79,80], which are structured on three main levels [81]: (1) educational spaces as a pedagogical element that promotes social learning, (2) an organizational proposal that defines the pedagogical framework of cohabitation and (3) the community as a learning structure [82], with special attention paid to the processing and management of the educational and academic data collected [83].

With the desire to continue advancing the development of a specific curriculum for educational robotics and computational thinking, La Salle Bonanova School (Barcelona, Catalonia) is entering the second year of the implementation of educational robotics and computational thinking as part of the curriculum at the secondary level. The secondary stage is distributed over four years of age as specified in the Government of Catalonia’s law [84], and the curriculum details the contents that students must learn, the skills they must achieve and the digital age they much reach by the end of the educational stage (based on the Curriculum of Secondary Education, Basic skills in the digital realm, Identification and deployment in compulsory secondary education of the Generalitat de Catalonia Law [85]).

The subject of technology is included in this curriculum, and therefore, the contents and competences are very well specified. In the case of computational thinking and educational robotics, there is no clear development of the curriculum, and certain freedom is left to each school to develop the skills and contents according to their own resources and the methodology they consider most appropriate. In the case of La Salle Bonanova, the structure and methodology for the inclusion of these two aspects were well-defined in the first year of implementation of the project [40]. In this first year, the bases were defined so that all the students who attended secondary education developed their own abilities and competences in computational thought and educational robotics, while STEAM subjects were strengthened.

The main lines that were defined in terms of content were: knowledge related to technology, computer science, coding and the use of information technologies. In terms of the methodology and transversal knowledge, learning was approached through contextualized activities and the development of skills in five areas: communication, collaboration and community building, context creation, creativity and conduct or behavior [54].

### 3.2. Educational Context

In the first year (a non-pandemic academic year), the first, second and third grades of secondary school (445 students in total) were introduced to educational robotics and computational thinking. This first contact with these disciplines developed in the corresponding role [40], and as a summary, it consisted of:15 sessions to introduce block-programming platforms such as Scratch™ and LEGO^®^ Mindstorms^®^ as basic tools for computational thinking and educational robotics.Work in two groups: where activities were or were not contextualized.Differentiation of sessions between one group and the other.Definition of the contents related to programming.Definition of the competences to be worked on, grouped into five areas.Analysis of the results obtained by the two groups.

In the second year of implementation, activities were still carried out in all the courses described above, but the focus was on project-based learning (*n* = 160 students, ages 13–14). Students had learned about robotics for the first time during the previous year, and we decided to delve into the basic aspects of PBL to enhance their computational thinking and knowledge of educational robotics, building on the knowledge of Scratch™ acquired in the previous academic year. 

### 3.3. Previous Knowledge Acquired

As in any other discipline in education, robotics and computational thinking’s teaching content is based on prior knowledge [67], and help and resources are provided to achieve the learning objectives [86]. To develop the full potential of the project, students learning on these courses built on the concepts they learned on the previous course during the Scratch™ activity [40]. Regardless of their group, the vast majority of students used block-programming platforms such as Scratch™ and gained knowledge about computational concepts. Our evaluation was performed on a scale of 1–10 (Scratch™ M = 5.31, SD = 1.04; Computational concepts M = 6.00, SD = 1.05).

The aspects we considered when evaluating their use of Scratch™ are detailed in Table 1, which were worked on and developed during the sessions in this research project. 

### 3.4. Previous Skills Development

In today’s education system, concepts are still important, but student development based on their skills and abilities is gaining momentum. As mentioned above, there is no specific robotics curriculum that develops the content, and in the same way, there is no curriculum to develop the skills. There are several jobs in the field of early childhood education [54] in which a series of capabilities are defined that could be described as generic without being specific to a curriculum. These five areas are the ones used in the first year of application of this project [29], and they are the ones that continue to be worked on and developed in this field. The competences are detailed in Table 2.

In the case of competency development, the assessment was performed following a Likert scale of 1 to 5. The results for each group expressed as averages were C1: 2.68/3.29, C2: 2.48/2.96, C3: 2.25/2.64, C4: 1.66/2.57 and C5: 3.73/3.95. The first number of each competency evaluated corresponds to the group in which it was assessed without contextualization of the activities, and the second corresponds to the group that worked with contextualized activities.

As in a previous study [40], there was a significant difference in competence between groups. This reaffirms the conclusions of the previous research on the value of using contextualized activities to improve students’ abilities and skills related to educational robotics and computational thinking. This conclusion last time gave value to the present study, encouraging us to follow an even more inclusive methodology and apply PBL to educational robotics and computational thinking.

Following the work of Brennan and Resnick [47], we related the four of the competences assessed to the CT perspectives, highlighting the relationships between the two, as shown in Table 3:

This relationship can be used in future research to establish connections between competences within the curricular framework, and can help to improve both the design of activities and their evaluation.

### 3.5. Methodology 

The work methodology used was project-based learning. The main aim of PBL is to improve the skills needed in the 21st century by following a series of guidelines and premises so that students are the drivers of their learning and the teacher is the guide or facilitator [26].

Students present the PBL outcome as if it were the final project of the Scape Room using the block-programming platform Scratch™. As this was the second year of use of this platform, they were building on previous knowledge, and therefore, it was not necessary to give a conceptual introduction to the contents of the platform. During the work sessions, the students developed the history and programming of the Scape Room, and the teacher served as a guide to dispel their doubts and support them to produce a final product.

In the PBL methodology, peer collaboration is very important, which involves sharing one’s knowledge for the benefit of the group, so that you reach a common goal [26].

Trial-and-error is also considered a very important factor. As this was a Scape Room, where before coding, it is necessary to create a thread and draw up a list of all the objects to be programmed (as specified in Section 3.5.2), the design may change during the development of the program. This fact is fully understood by the teachers on the project, but not most students. It is precisely at the point when substantial changes to the previous design are required for the material realization of the programming that difficulties will appear and 21st-century skills will be developed, such as communication, respect for others, teamwork and problem-solving [26], all while learning a computational language and programming.

#### 3.5.1. Criteria for the Composition of Working Groups

All second-grade students were included in this study. A total of 160 students participated, comprising five classes of 32 students each (60% boys, 40% girls). The age range was between 13 and 14 years.

The classroom work was based on groups of four, encouraging peer collaboration and cooperative learning on the subject of technology. This cooperative work sought to help students better identify concepts, perform problem analysis and foster peer relationships [67].

In a previous study we carried out at La Salle Bonanova School [40], educational robotics and computational thinking were introduced. As already described in this study, two different working methodologies applied, producing a significant difference in the level of competences gained between members of one group and the other, though the assimilation of the computational concepts learned was homogeneous for the two groups.

In this study, new criteria were proposed: the working groups comprised equal numbers of students from each of the two groups in the previous study; beyond meeting that criterion, they were randomly selected. In this study, 16 students at once (four groups of four students) participated in two-hour sessions.

#### 3.5.2. Development of the Sessions

The PBL that was developed was a Scratch Room, where through the Scratch™ platform, a Scape Room was designed. A plan for the sessions is shown in Table 4.

The first step students take is a storyboard, designing their Scape Room in scenes. Afterward, as a first step to start using the Scratch^TM^ platform, they must list all the objects, scenarios, etc., that they will need to program. In these two sessions, design thinking is the basis for later being able to achieve good programming. Already, in these first steps, trial-and-error is important since some elements or actions from the storyboard may not be possible to perform with Scratch^TM^, requiring a rethink and redesign of the proposal.

From the third to the sixth sessions, elements of computational thinking and educational robotics come into play, such as problem-solving, trial-and-error and using elements of the Scratch^TM^ platform to develop a tangible product.

Elements already mentioned in this paper, based on the work of Brennan and Resnick [47], are the basis for the program’s cracking process. Computational concepts include the use of sequences, loops, events, parallel programming, conditionals and mathematical operators. As computational practices, students relay iterations and incremental programming, and they will be constantly testing and debugging, as they have to test and improve their programming. As they progress in their programming, especially from the sixth session, the programming becomes more complex, and therefore, students have to do some work of abstraction and modularization. The oral presentation and the test they do with their Scape Room during the sixth session serve to highlight possible improvements and mistakes, allowing them to develop computational perspectives. To progress, they must make use of connectivity, express their ideas and question aspects of the whole creative process.

During the development of the sessions, enhanced by the fact that they work cooperatively, students develop aspects of computational thinking that can be extracted and applied to any robotic platform as these are based on design thinking and problem-solving, extending beyond knowledge of the platform and its specific elements.

#### 3.5.3. Methods and Tools of Analysis

The authors used two tools to collect data:

The first corresponded to data related to the development of the competences and skills, which were collected through Rubric 1. In each work session, we observed how the students had worked, taking into account all the competences. The results are attached at the end of this document via a link and are summarized in Table 5.

Data on the CT concepts demonstrated when using Scratch^TM^ platform were then collected using Rubric 2. As shown in Figure 1, these assessments were made in the middle of the project, in session 6, and at the end of the project, in session 10. We assessed the use of certain CT concepts, the degree of complexity, the understanding on the part of the students and the good operation of the programming. The results are attached at the end of this document via a link and are summarized in Table 6.

Analyses of these data, via descriptive statistics, were treated independently, so the means and corresponding standard deviations of each item were obtained separately, both for the competences and the CT concepts.

One of the impediments to being able to carry out a more in-depth analysis was the short duration of the sessions and as it was the first time the rubrics had been used. Although the authors are aware of this limitation, we consider that the results are sufficiently significant to be considered as representing the students’ acquisition of concepts within the training framework.

## 4. Results

During the sessions, different aspects were evaluated, such as the skills gained and contents learned relating to computational thinking and programming with the Scratch™ platform.

### 4.1. Assessment of Students’ Skills (R1)

In the assessment of the PBL and the acquisition of skills by students, to answer research question 1, three scores were awarded: 

The first was based on the direct observation of each student with a rubric, either for their interaction with the group members or their individual development during the sessions; in this way, assessments of competences C1, C2, C3, C4 and C5 were obtained. Competence C5 is not typical of CT (as shown in Table 3) but instead of cooperative work; the authors considered it useful to evaluate a student’s work among equals.

The second point that was valued was the oral presentation and how they developed their project idea and justified the decisions made. 

The third point of assessment was the final project and the involvement of the group members themselves through a co-evaluation.

Although three aspects were assessed, the authors focused their attention on the competences and not so much on the oral presentation or co-evaluation. The reason for this was that one of the aims of developing project-based learning is to observe if the development of certain competences related to computational thinking is favored over the others. As described above, these competences are those defined in the first year of implementation and correspond to Positive Technological Development (PTD) [54]. It has already been mentioned that no special grouping was done or any different methodology applied. We only ensured that the groups were homogenized in relation to the first year of application of the project. The averages of the five areas for all students, on a Likert scale (1–5), are shown in Table 5. 

### 4.2. Assessment of Computational Thinking Concepts (R2)

To answer research question 1, the authors took into account the results obtained from the assessment of the use of the Scratch^TM^ platform.

Our findings on the CT concepts students demonstrated when using the Scratch^TM^ platform (follows the items listed in Table 1), as assessed on a scale of 0 to 10 by quantifying the functional blocks corresponding to the CT concepts, are shown as the overall mean and standard deviation in Table 6.

We then checked whether the results of the study carried out in the previous year correlated with the new results. In the last study, the group that did not follow an active methodology and worked on robotics and computational thinking in a classical way obtained inferior results to the group in which a methodology was applied where the activities were contextualized and competences and the learning process were accounted for.

Table 7 shows how the overall results for the second year competences were very similar to or higher than those of the second group in the first year. This was interesting since, in the second year, the groups were mixed and formed by students who had followed, in the previous year, different methodologies. Nonetheless, despite the difference in the degree of acquisition of skills, the methodology used promoted rapid and meaningful learning of the key aspects of STEAM subjects, as well as educational robotics and computational thinking.

The other fundamental aspect of this work was the evaluation of CT concepts. Following our assessment of the acquisition and use of CT knowledge, as demonstrated when using the Scratch™ platform, Table 6 shows the mean progress in the second year of the project working with PBL (M = 6.36, SD = 0.9). The results contrast with those obtained in the first year (M = 5.66, SD = 1.03) when the work environment was contextualized activities [40].

## 5. Discussion

Based on the data obtained, and considering the two research questions, we will now discuss the implementation of an active methodology such as PBL in deploying the NCA project.

### 5.1. Research Question R1

As stated in the pedagogical principles developed in the NCA project [77,78,79,80], one of the aspects that emerge above others is the development of the person. It is for this reason that the present study took into account the skills and abilities most needed in the 21st century. 

In this sense, one of the possible limitations of this study, integrated into the NCA project, was the complexity of assessing non-formative aspects, such as attitudes and skills. However, we consider it essential to have open projects and active methodologies, such as project-based learning, to give flexibility to the design of projects and activities, meaning each project can be adapted to the profiles of different students.

In response to research question 1, we consider that using project-based learning encouraged and reaffirmed work to develop competences. The data obtained and set out in Table 5, when compared with previous studies [40], demonstrate improvements in the acquisition of competences, as shown in Table 7.

The program designed to develop skills in this study will contribute to other studies in other educational stages [23]. It was based on the competences developed in previous work with block-based programming platforms [47], as shown in Table 3. Yet, the nature of the concept of a skill or competence in itself is already a limitation. 

Thus, one of the characteristics to be accounted for, and where we can improve objectivity, is the definition and evaluation of the competence items. It remains for anyone to establish competences specifically for the secondary stage curriculum where these are accurately described.

### 5.2. Research Question R2

We sought to answer research question 2 based on the definitions and items given in Table 1. These concepts were worked on using blockchain-based programming platforms such as Scratch^TM^, and therefore, were specified based on such platforms’ characteristics. Accordingly, when developing knowledge of CT concepts, one of the limitations for the students we assessed was the use of a single programming resource. To improve on this, future work will be aimed at designing learning activities decoupled from a specific technology. However, this study intended to deepen the students’ knowledge of Scratch^TM^, and therefore, this limitation is not considered to have prevented us from being able to answer the corresponding research question.

Another aim of this study was to analyze the impact that the PBL methodology had on activities designed to teach CT concepts and see how the motivation and/or involvement of students in carrying out more complex projects was impacted. The data shown in Table 6 (M = 6.36, SD = 0.9), and their evolution when compared with previous studies (M = 5.66, SD = 1.03), affirm that this type of activity is optimum for working on computational thinking concepts.

In general, the implementation of this type of study in a context such as the NCA addresses the challenge of how to reproduce, replicate, scale and adapt a type of activity for all schools, groups, levels and types of students.

Although our conclusions are optimal and encourage us to continue with this project, the truth is that using the same types of activities and projects reduces the uniqueness of student learning and limits the creativity of teachers. Therefore, one of the major limitations that may arise is the homogenization of learning activities related to the concepts of computational thinking.

Nonetheless, the authors propose to establish an approach that is in accordance with the principles of the NCA project: use objective evaluation tools (rubrics), define the contents and concepts related to computational thinking and relate them to the official curriculum.

## 6. Conclusions

Educational robotics, science and technical-scientific practice, in general, have become increasingly common in personal, industrial and educational contexts [87,88].

In the educational environment, larger-scale schools or school institutions are evoking an increasingly integrated use of technological resources such as robotic platforms or virtual programming environments. The work presented in this paper, and other previous studies related to STEAM subjects’ [8,9,10], computational thinking [40,47] and project-based learning [25,69], have brought to light a series of elements within the educational field to be considered.

First, it should be noted that an active methodology such as project-based learning can increase the performance and motivation of students, and thanks to these, they become the protagonist of their learning and generate a physical product because of their learning.

Students who have participated in a second year of using block-based programming platforms within a PBL environment improve their skills gained and acquisition of knowledge of computational thinking and the use of educational robotics platforms. This improvement in the computational thinking area covers virtually every aspect—concepts, practices and perspectives—and relates to the competences previously defined in this study, such as communication between classmates, creativity, collaboration, community building and context creation, as seen in the results presented in Table 7.

In relation to this last statement, the authors conclude that the relationship between CT practices and competences, as indicated in Table 3 as a good first approximation, represents how students interact with each other within programming environments. 

Another positive aspect to highlight in the implementation of this study is the possibility of repeating and graduating. This type of PBL can be reproduced within the same school in other STEAM subjects and also in other schools. The possibility of repeating and graduating hinges on adapting the timing of the program and the composition of the working groups, while adapting the level according to the previous knowledge and educational stage, but still maintaining the same curriculum and objectives of learning CT practice and concepts.

Finally, we conclude that integrating cross-curricular computational thinking into STEAM subjects is an added value for schools, and one that is increasingly necessary.

From the data obtained, it can be seen that for La Salle Bonanova school, promoting students’ development of the skills needed in the 21st century and their learning and development of computational thinking are important aims and set the direction for the school’s technological educational project. That project can be consolidated by encouraging learning in both ER and CT and designing a strategy to integrate those into the curriculum of STEAM subjects in the secondary stage.

This research was part of the ‘New Learning Context’ (NCA) launched by the schools of La Salle Educational Mission Assembly (AMEL 2018) [77,78]. La Salle Bonanova is one of the biggest schools in Spain and a pioneer in both research and teaching STEAM subjects. The development of this project as it continues is part of a pilot test to lay the foundations for future actions in other schools within the institution. The study was carried out within a school year and without modifying the school structure, curriculum or schedule.

Based on the results and conclusions of this study, the authors consider that the methodology (PBL) and type of activity align with the principles set by the institution. The number of students in the sample was significant as it was a whole stage (*n* = 160) and no students were excluded. Therefore, the sample was important in terms of objectivity for representing students’ level of attainment in an ordinary year of secondary school.

In parallel with the development and implementation of the NCA, the two universities involved are developing a training plan for teachers, both in STEAM and other subjects, as the mission is based on five pedagogical principles [79,80], and for these to be effective, it is necessary to train all teachers who part of the schools involved. The implementation process of the NCA has several phases and will be carried out according to various parameters of each school. Through these projects, the institution advances and it will soon be able to establish activities and methodologies that can be reproduced and adapted to all schools and levels.

## Figures and Tables

**Figure 1 sensors-22-03746-f001:**
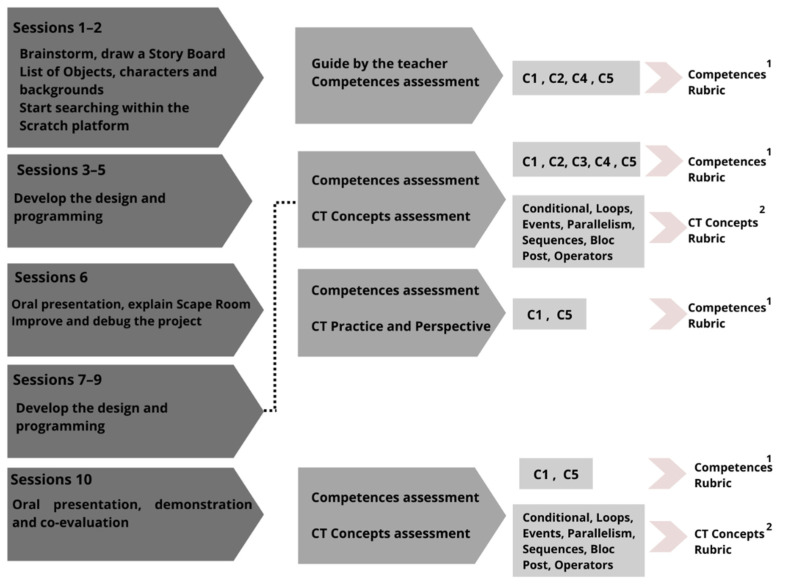
Flowchart of the sessions, assessment concepts and tools used. 1: Mean and standard deviation for each student and competence. 2: Mean and standard deviation for each student and CT concept.

**Table 1 sensors-22-03746-t001:** Items evaluated on the Scratch™ platform.

Type	Description
Conditional	Understand the concept of conditional structures
	Use different types of conditional structures
Loops	Understand the concept of a loop iteration
	Use loops within the structure of the game
Events (Objects)	Use various objects
	Import objects from outside of Scratch™
	Motion control is done in several ways
	Use of objects follows criteria established
Events (Scenario or Dresses)	Use different scenarios
	Make changes to objects’ dresses
Parallelism	Implement several sequences in the same object Different actions for each object
Bloc Posts	Use the blog post to give orders to objects
Sequences (Text)	Use language structures
	Dialogues appear
Operators	Use variables to make a counter increase
	Use variables to make a counter decrease
	Conditions certain actions variables
Events (Music/Sounds)	Use music blog
	Use varied sounds
	Use block sound conditioning in another action

**Table 2 sensors-22-03746-t002:** Aspects of competences evaluated.

Type	Description
C1 Communication	C1.1 Exchange of ideas among group members
	C1.2 Expression of ideas and debating them
	C1.3 Demand for teacher support and benefit to project
C2 Collaboration and Community Building	C2.1 Helps peer group
	C2.2 Individual contributions make the group advance
	C2.3 Different work roles/task diversity
C3 Context Creation	C3.1 Activity follows a designed structure
	C3.2 Analysis of errors in the process
	C3.3 Justification of the solution
	C3.4 Writes the process of the solution to the challenge
C4 Creativity	C4.1 Holds initiative to make further steps in programs
	C4.2 Use of various elements outside environment of platform
	C4.3 Application of concepts from other disciplines
C5 Conduct	C5.1 Concentration activity
	C5.2 Following the rules of the classroom
	C5.3 Responsible use of the material
	C5.4 Behaves well with classmates and teacher
	C5.5 Motivated to complete activity

**Table 3 sensors-22-03746-t003:** Relationships between CT perspectives and competences.

Computational Perspectives [47]	Competences [40,54]
Expressing	C1 Communication
Questioning	C2 Collaboration and community building
Connecting	C3 Context creation C4 Creativity

**Table 4 sensors-22-03746-t004:** Session plan.

Session(s)	Developed
1	Groups of four students are formed and an explanation is given for a Scape Room. Students are asked to brainstorm a possible Scape Room. A worksheet is handed out for each group to start drawing a storyboard of the chosen Scape Room.
2	The storyboard is finished and a list is made of all the possible objects, characters and backgrounds needed. The students start searching within the Scratch^TM^ platform for the objects and backgrounds that are needed; otherwise, they can search websites.
3, 4, 5	During these three sessions, students develop the design, and above all, the programming of their Scratch Room. They must define the complexity of their project and thus establish their learning.
6	An oral presentation is given on each project where students explain the operation of their Scratch Room. It is a checkpoint to improve and debug the project. In the Q&A, comments and observations are offered by the rest of the groups.
7, 8, 9	During these three sessions, students continue to develop their project and improve it based on the observations from the previous session.
10	Oral presentation, demonstration and co-evaluation of the final project.

**Table 5 sensors-22-03746-t005:** Mean and standard deviation of competences.

Competences									
C1		C2		C3		C4		C5	
Mean	SD	Mean	SD	Mean	SD	Mean	SD	Mean	SD
3.22	0.62	2.98	0.72	3.17	0.71	3.19	0.70	3.98	0.66

**Table 6 sensors-22-03746-t006:** Mean and standard deviation of CT concepts demonstrated when using Scratch^TM^.

CT Concepts	Scratch™	
	Mean	SD
	6.36	0.9

**Table 7 sensors-22-03746-t007:** Comparison of the first year vs. second year competent areas.

Competences	First Year					Second Year	
	First Group		Second Group		First and Second Mixed Group		
	Mean	SD	Mean	SD	Mean	Mean	SD
C1	2.68	0.50	3.29	0.69	2.98	3.22	0.62
C2	2.48	0.64	2.96	0.74	2.72	2.98	0.72
C3	2.25	0.77	2.64	0.80	2.45	3.17	0.71
C4	1.66	0.65	2.57	0.80	2.12	3.19	0.70
C5	3.73	0.86	3.95	0.77	3.84	3.98	0.66

## Data Availability

The data obtained from the study and endorsed by La Salle Bonanova School and La Salle-Ramon Llull University are available at: Computational Thinking & Educational Robotics integrated into PBL.

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
