# Peer review of "Computational Thinking and Educational Robotics Integrated into Project-Based Learning"

_sensors, 2022, doi:10.3390/s22103746_

Round 1

Reviewer 1 Report

Abstract

The content of the abstract has written too much about the research background and research purpose of this study. While the process and conclusion of this study are not described in detail.

Introduction

First, the introduction of this article describes too many aspects. However, it seems too long for the reader and it is recommended that this section be rewritten, particularly around the research objectives and motivations of this study. In addition, why TPACK was mentioned in this study?

Second, there is no clear research question proposed in the manuscript. It can be added in the last part of introduction.

Third, section 1.1 can be moved into section 3.

Fourth, the literature has been extensively studied. However, considering how prolific the field is, it has to extend the bibliography with significant, current references (up to 2022) and show that the problem is still relevant.

Literature review

First, the comprehensive content of computational thinking was shown in 2.1. There are many explanations of computational thinking structure in previous studies. Why does this manuscript only list the structures of computational thinking proposed by Brennan and Resnick? What was the reason for choosing this model?

Second, Scratch is confusing as a separate section in the literature review. why does this study emphasize Scratch?

Third, the literature review should also include an overview of existing research on educational robotics.

Course Design and Methodology

This study discusses computational thinking and educational robotics integrated into project-based learning. But the manuscript does not specifically address how design thinking and educational robotics are integrated into project learning activities.

Instrumental tools that were used to examine the difference between the experimental group and control group were not introduced.

The research process of this study is not clear enough. The procedure of the experiment can be depicted in a flow chart.

Other

Effect size needs to be calculated.

The discussion and limitations of this study need to be added to the article.

Reviewer 2 Report

The first part of the paper gives an overview of relevant literature, the second part describes an experience of an educational program (a follow-up project) with 160 students in one  secondary school to integrate Computational Thinking and Educational  Robotics by using Project-Based Learning. The used methodology can also be implemented in other schools.

Strengths: The authors demonstrate good awareness of related literature, there are 88 references. The discussion and the results are illustrated with 7 tables. 

Comments: Abstract is too general, it is hard to understand the aim and the results of the study. Acronyms should be avoided in Abstract.

Future work is not quite clear. Do you plan to implement the methodology in other schools of Spain? Are the teachers prepared?

Row 286: TC practices?

Row 402: the evaluation was performed on a scale of [...]?

Round 2

Reviewer 1 Report

There are still many confusing aspects of the research design and the style in which the paper was written.
1. TPACK was introduced in section 1. It was originally proposed for teachers. From the perspective of an educational experimental design, TPACK was neither an independent nor a dependent variable in this study. Is TPACK a training model in this study? In this way, the content of TPACK seems redundant. In particular, the research questions have little relevance to TPACK.
2. The authors added two research questions in the last paragraph of section 1: Would working through project-based learning improve students' skills? will using a visual programming platform facilitate the acquisition of programming concepts?
It will be easier for readers to read the manuscript if the author reports the Results and Discussion of these two research questions, respectively. That is, the results are divided into two sections, respectively for question 1 and question 2.
3. The manuscript should have a specific section describing the analysis methods used in this study to respond to the two research questions.
4. The format of section 3.2 is problematic.

Round 3

Reviewer 1 Report

The revised manuscript can answer the comments. However, the language should be proofread and the plagiarism should be detected.